# Influence of the Fatty Acid Metabolism on the Mode of Action of a Cisplatin(IV) Complex with Phenylbutyrate as Axial Ligands

**DOI:** 10.3390/pharmaceutics15020677

**Published:** 2023-02-16

**Authors:** Theresa Mendrina, Isabella Poetsch, Hemma Schueffl, Dina Baier, Christine Pirker, Alexander Ries, Bernhard K. Keppler, Christian R. Kowol, Dan Gibson, Michael Grusch, Walter Berger, Petra Heffeter

**Affiliations:** 1Center for Cancer Research and Comprehensive Cancer Center, Medical University of Vienna, Borschkegasse 8a, 1090 Vienna, Austria; 2Faculty of Chemistry, Institute of Inorganic Chemistry, University of Vienna, Waehringer Strasse 42, 1090 Vienna, Austria; 3Research Cluster “Translational Cancer Therapy Research”, 1090 Vienna, Austria; 4Institute for Drug Research, School of Pharmacy, The Hebrew University, Jerusalem 91120, Israel

**Keywords:** platinum(IV), phenylbutyrate, cisplatin, prodrug, lipid metabolism, Warburg effect

## Abstract

For a variety of cancer types, platinum compounds are still among the best treatment options. However, their application is limited by side effects and drug resistance. Consequently, multi-targeted platinum(IV) prodrugs that target specific traits of the malignant tissue are interesting new candidates. Recently, cisPt(PhB)_2_ was synthesized which, upon reduction in the malignant tissue, releases phenylbutyrate (PhB), a metabolically active fatty acid analog, in addition to cisplatin. In this study, we in-depth investigated the anticancer properties of this new complex in cell culture and in mouse allograft experiments. CisPt(PhB)_2_ showed a distinctly improved anticancer activity compared to cisplatin as well as to PhB alone and was able to overcome various frequently occurring drug resistance mechanisms. Furthermore, we observed that differences in the cellular fatty acid metabolism and mitochondrial activity distinctly impacted the drug’s mode of action. Subsequent analyses revealed that “Warburg-like” cells, which are characterized by deficient mitochondrial function and fatty acid catabolism, are less capable of coping with cisPt(PhB)_2_ leading to rapid induction of a non-apoptotic form of cell death. Summarizing, cisPt(PhB)_2_ is a new orally applicable platinum(IV) prodrug with promising activity especially against cisplatin-resistant cancer cells with “Warburg-like” properties.

## 1. Introduction

Platinum compounds still play a very prominent role in current standard anticancer therapy regimens. Cisplatin, discovered in the 1960s by Rosenberg et al., was the first of three platinum compounds to ever achieve worldwide approval by regulators [1]. Cisplatin is administered intravenously and enters the cells from the blood stream either passively or actively via copper transporters such as CTR1 [2,3]. Intracellularly, cisplatin is hydrolyzed and subsequently induces cell death by crosslinking DNA. Even though cancer cells are more sensitive to cisplatin in comparison to cells from healthy tissue, dose-limiting side effects and occurrence of resistance reduce therapy success. To overcome these restraints and generally improve anticancer efficacy, several strategies have been investigated. Among them, the prodrug concept of platinum(IV) compounds has shown promising results. The higher oxidation state of the platinum center increases the compound’s kinetic inertness thereby reducing possible side effects in normal tissues [4,5,6,7]. Moreover, two additional (bioactive) axial ligands can be introduced, e.g., altering lipophilicity, pharmacokinetics, or generating multi-targeted complexes. In the malignant tissue, the characteristic reductive environment is supposed to activate the platinum(IV) complex to its cytotoxic platinum(II) counterpart and release the axial ligands [8]. A few platinum-based prodrugs have already been investigated in clinical trials. One example is satraplatin, the first orally available platinum(IV) prodrug, which has been investigated in multiple clinical trials since 2005 [9]. Of note, satraplatin successfully reached clinical trial phase III (NCT00069745), but ultimately failed approval due to a lack of superior efficiency for overall survival compared to standard therapy. Nevertheless, utilizing the paradigm of platinum(IV) prodrugs is an effective way to improve anticancer activity and to overcome resistance mechanisms [10,11], for example, by attaching synergistic compounds to the axial position(s). Furthermore, 4-phenylbutyrate (PhB, 4-PBA), clinically approved for the treatment of urea cycle disorders, has recently attracted attention as an anticancer compound due to its promising synergistic activity with DNA-binding drugs [12,13]. In fact, PhB is a fatty acid analog with a wide range of applications and suggested modes of actions [14]. On the one hand, PhB is a chemical chaperon, stabilizing protein conformation and, thus, one of the most frequently used endoplasmic reticulum (ER) stress inhibitors [15,16,17]. On the other hand, PhB influences cellular metabolism by binding to coenzyme A (CoA) via thiol adduct formation [18] as well as inhibiting histone deacetylase (HDAC) [19]. Moreover, PhB is able to directly inhibit pyruvate dehydrogenase kinase 1 (PDK1) in the mitochondria [20]. Even though many pathways are not yet fully understood, the strong synergism with cisplatin [21,22] prompted the development of “dual-action” cisplatin-based platinum(IV) prodrugs [8,23]. For example, cisPt(PhB)_2_, which carries two axial PhB residues, showed encouraging preliminary results in cell culture experiments. The aim of this study was to further characterize the biological activity of this compound in (cisplatin-resistant) cancer cells in vitro and in vivo. Our collected data indicate that the new platinum complex is especially interesting for the treatment of cancer cells with a pronounced “Warburg”-like metabolic phenotype.

## 2. Materials and Methods

### 2.1. Chemicals and Reagents

CisPt(PhB)_2_ was prepared as previously published [8], dissolved in dimethyl sulfoxide (DMSO) to obtain a stock solution of 10 mM, and then aliquoted and kept at −20 °C until further analysis. Cisplatin was purchased from LC laboratories (Woburn, MA, USA). For cell culture use, cisplatin was dissolved in dimethylformamide (DMF) to obtain a stock solution of 11 mM. The solution was then diluted in serum-free RPMI-1640 medium to a concentration of 5 mM. 4-phenylbutyric acid (Sigma Aldrich, St. Louis, MI, USA) was freshly dissolved in double-distilled H_2_O to obtain a stock concentration of 20 mM. Triacsin C (Merck, Darmstadt, Germany, 10 mM stock), perhexiline (Merck, Darmstadt, Germany, 10 mM stock), etomoxir (Adooq Bioscience, Irvine, CA, USA, 100 mM), and 5,5,6,6-tetrachloro-1,1,3,3-tetraethylbenzimidazol-carbocyanine iodide (JC-1, Enzo Life Sciences, New York, NY, USA, 1 mg/mL) stock solutions were prepared in DMSO and stored at −20 °C until analysis.

### 2.2. Cell Culture

The cell lines, their sources, and specific growth medium are summarized in Appendix A. TP53 status for A2780, Capan-1, PANC-1, and MDA-MB231 cells were extracted from the IARC database, Aug 2021; for VM-1, TP53 status was assessed in the course of the precision medicine platform MONDTI [24]. All media were supplemented with 10% fetal calf serum (FCS, PAA, Linz, Austria). Cells were kept in a humidified atmosphere at 37 °C and 5% CO_2_. Cultures were checked for mycoplasma contamination before use. Cisplatin-resistant P31 and A2780 cells were selected every week with 4 and 1 µM cisplatin, respectively.

### 2.3. Cytotoxicity Assays

Cells were seeded in 96-well plates at 3500–6000 cells/well depending on the proliferation rate of the respective cell line. On the next day, the cells were treated with increasing concentrations of the compounds or their combinations for 72 h. Cell viability was determined using the 3-(4,5-dimethylthiazol-2-yl)-2,5-diphenyltetrazolium bromide (MTT)-based vitality assay (EZ4U; Biomedica, Vienna, Austria) following the manufacturer’s recommendations. To calculate the IC_50_ values (concentration that induces a reduction in cell number to 50%), full dose–response curves were generated using GraphPad Prism (GraphPad Software, San Diego, CA, USA, version 8.0.1 for Windows).

### 2.4. ICP-MS Measurements of Cell Uptake

Cells were seeded in triplicates in 6-well plates (Starlab, Hamburg, Germany) at concentrations to reach 80% confluence after 24 h. Cells and blank wells containing no cells were treated with 5 or 10 µM of the compounds for 2 h. Subsequently, wells were washed twice with phosphate-buffered saline (PBS) and incubated with 67–69% nitric acid (VWR, Darmstadt, Germany) for 1 h. The lysates were further diluted 1:20 with H_2_O and analyzed using inductively coupled plasma-mass spectrometry (ICP-MS) for their platinum content. The cell number was calculated from duplicates, and platinum concentration was normalized to the cell number. Samples were measured using an Agilent 7800 ICP-QMS instrument (Agilent Technologies, Tokyo, Japan) equipped with an Agilent SPS 4 autosampler (Agilent Technologies, Tokyo, Japan) and a MicroMist nebulizer at a sample uptake rate of approx. 0.2 mL/min. The Agilent MassHunter software package (Workstation Software, Version C.01.04, 2018, Agilent, Santa Clara, CA, USA) was used for data evaluation. All of the measured samples were blank-corrected. Elemental standard solutions were purchased from Labkings (Hilversum, The Netherlands). The instrument was tuned on a daily basis.

### 2.5. Animals

Eight- to twelve-week-old BALB/c (Envigo, San Pietro Al Natisone, Italy) and C57BL/6 mice (Janvier Labs, Le Genest-Saint-Isle, France) were kept in a pathogen-free environment with a 12 h light–dark cycle with ad libitum access to food and water. Every procedure was performed under sterile conditions. This study was conducted according to the guidelines of the Declaration of Helsinki, approved by the Ethics Committee for the Care and Use of Laboratory Animals at the Medical University Vienna (proposal number BMWF-66.009/0084-II/3b/2013), and performed according to the guidelines from the Austrian Animal Science Association and from the Federation of European Laboratory Animal Science Associations (FELASA).

### 2.6. Anticancer Activity In Vivo

Murine CT26 (5 × 10^5^ cells) or B16 cells (1 × 10^5^ cells) were injected subcutaneously into the right flank of male BALB/c or C57BL/6 mice, respectively. Body weight was measured daily. Animals were treated with cisPt(PhB)_2_ (20 mg/kg, dissolved in 10% DMSO, p.o.). The concentration of cisPt(PhB)_2_ was chosen based on previously unpublished toxicity studies (personal communication D. Gibson). Tumor size was measured daily using a caliper and tumor volume was calculated with the formula: (length × width^2^)/2. Animals were sacrificed if tumors were ulcerated or if the tumor length exceeded 2 cm.

### 2.7. Live-Cell Microscopy

Cells were seeded in a µ slide (8-well glass bottom, ibidi, Gräfelfing, Germany) at a cell number of 4 × 10^4^ cells/well and allowed to recover for 24 h. In the case of the lipid droplet analysis, bodipy™ 493/503 (Thermo Fisher, Waltham, MA USA) was added 15 min prior to treatment at a concentration of 0.5 µM to the supernatant. Cells were treated with the compounds for 72 h and two images per well (bright field and FITC-channel) were taken every 20 min using a Nikon Eclipse Ti with a 20X (Super Plan Fluor NA 0.45 Ph1) objective in an OkoLAB Incubation Box (5% CO_2_, 37 °C, passive humidifier) with a PCS sCMOS monochrome camera 4.2 MPxl. Images at specific time points (0, 6, 12, 18, 24 h) were analyzed using ImageJ software. The image background was subtracted, the threshold was applied, and the integrated density as the area occupied by cells was quantified. For the analysis of migration, the cells were manually tracked over 24 h using ImageJ, and the coordinates for each individual cell and time point were obtained. The DiPer migration tool for Microsoft Excel was used to calculate the speed of cell migration and generate plots of origin for each cell to depict individual movements over time [25].

### 2.8. Cell Cycle Analysis

Cells were seeded in 6-well plates (Starlab, Hamburg, Germany) at concentrations to reach 70% confluence on the next day. Cells were then treated for 24 h and collected using trypsinization. Cell pellets were resuspended in 100 µL 0.9% NaCl and added dropwise to 1.8 mL 70% ethanol for fixation (at least 1 h at −20 °C). Fixated cells were centrifuged and incubated with 100 µg/mL RNase A (Merck, Darmstadt, Germany) for 30 min at 37 °C. Subsequently, nuclei were stained with 5 µg/mL propidium iodide (PI, Merck, Darmstadt, Germany) for 30 min at 4 °C and analyzed using flow cytometry (LSRFortessa^TM^ X-20 Cell Analyzer, BD Biosciences, Franklin Lakes, NJ, USA). Quantification was performed using FlowJo_V10 software (Becton, Dickinson and Company, Franklin Lakes, NJ, USA).

### 2.9. Cell Death Analysis

CT26 or B16 cells were seeded in 6-well plates (5 × 10^5^ cells in 1 mL per well) and left to recover overnight. The next day, cells were treated with 5 µM of cisPt(PhB)_2_ for different time points. The supernatant and trypsinized cells were collected. As a control for dead cells, cells were incubated on a heating block (60 °C) for 30 min. The other samples were centrifuged with 300× *g* for 5 min and cells were stained with annexin V (AV, 1:50) and PI (1:50) in annexin V-binding buffer (ABB) (10 mM HEPES, 140 mM NaCl, 2.5 mM CaCl_2_) for 15 min at room temperature in the dark. The samples were diluted with 200 µL of ABB and directly measured at 530 and 610 nm using flow cytometry (LSRFortessa^TM^ X-20 Cell Analyzer, BD Biosciences, Franklin Lakes, NJ, USA).

### 2.10. Nucleus Staining

Cells were seeded in 6-well plates and allowed to recover overnight. The next day, cells were treated with the compounds for 24 h and collected using trypsinization. The cell suspension was transferred to a slide using a cytocentrifuge (Cytospin^TM^ 4, Thermo Fisher Scientific, Schwerte, Germany) at 1000 rpm for 5 min. Samples were fixated in an acetone:methanol mixture (1:1) for 10 min at −20 °C and mounted in VectaShield with 4′,6-diamidino-2-phenylindole (DAPI) (VECH-1200, Szabo Scandic, Vienna, Austria). Samples were analyzed using confocal microscopy (LSM700, Zeiss, Oberkochen, Germany). For each condition, five images were taken with a 63X Plan-Apochromat (NA 1.4 Oil DIC) objective (zoom 0.5, 203.2 × 203.2 µm) and nuclear morphology was analyzed and quantified under blinded conditions using ImageJ 1.53 t (Java 1.8.0_345 (64-bit), cell counter) and the particle analysis plugin.

### 2.11. Albumin Uptake

CT26 or B16 cells were seeded in 6-well plates (5 × 10^5^ cells in 1 mL per well) and left to recover for 24 h. FITC-conjugated bovine serum albumin (10 µM, A9771, Sigma Aldrich) was diluted with serum-free RPMI medium and added to the cells. After 3 h, the cells were harvested using trypsinization, diluted in PBS, and then fluorescence was measured at 530 nm using flow cytometry (LSRFortessa^TM^ X-20 Cell Analyzer, BD Biosciences, Franklin Lakes, NJ, USA).

### 2.12. Cellular Albumin Uptake

Cells were seeded (1.2 × 10^4^ cells/well) on 8-well chamber slides (Falcon™, Corning Brand, USA) in growth medium with 10% FCS and allowed to recover for 48 h. To determine the cellular uptake of albumin, the cells were treated with 10 µM FITC-labeled albumin (dissolved in serum-free RPMI medium) for 3 h and subsequently fixated using a solution of 4% paraformaldehyde dissolved in PBS with the pH adjusted to 7.4 (Merck, Darmstadt, Germany) for 15 min at room temperature. Spots were washed 4-timeswith PBS. Additionally, a staining solution of 0.3% DAPI and 0.45% rhodamine-labeled wheat germ agglutinin (WGA) (Vector laboratories, Newark, CA, USA) was added to stain cell nuclei and cell membranes, respectively. Next, the slide was mounted with non-hardening mounting medium (Vectashield^®^ Mounting Media, Vector laboratories, Newark, CA, USA), and analyzed using fluorescence measurement with a confocal microscope (Zeiss, Oberkochen, Germany) and image processing using ZEN lite software (Zeiss, Oberkochen, Germany).

### 2.13. Cellular Respiration Experiments

Cells were seeded into 96-well plates (XFe96/XF Pro Cell Culture Microplates, Agilent, USA) at a cell density of 2 × 10^4^ cells/well in 80 μL of cell culture medium supplemented with 10% FCS and cultured overnight. Cells were treated with 5 µM of cisPt(PhB)_2_ or solvent 4 h prior to measurement. The Seahorse Mito Stress Test (Seahorse XFp Cell Mito Stress Test Kit, Agilent, USA), with and without etomoxir, as well as the Glycolytic Rate Assay (Seahorse XFp Glycolytic Rate Assay Kit, Agilent, USA) were used for the measurement of the extracellular oxygen consumption rate (OCR) and extracellular acidification rate (ECAR). Assays were performed according to manufacturer’s recommendations. After the incubation period, the medium was replaced with Seahorse XF DMEM assay medium (pH 7.4, Agilent, Santa Clara, CA, USA) supplemented with 10 mM glucose, 2 mM glutamine, as well as 1 mM pyruvate, and incubated for 1 h in a CO_2_-free incubator at 37 °C. The kit reagents were sequentially added from the injection ports of the sensor cartridges (XFe96/XF Pro sensor cartridges, Agilent, USA) to a final concentration of oligomycin 1.5 μM, FCCP 1 μM, rotenone/antimycin A 0.5 μM, and etomoxir 4 µM, in the case of the Mito Stress Test, and of 2-deoxyglucose 50 mM and rotenone/antimycin A 0.5 µM, in the case of the Glycolytic Rate Assay. For quantification of cell numbers, 4 μM Hoechst 33258 (1 mg/mL in PBS, pH 7.4) was added. Following Seahorse analyses, cells were imaged, and Hoechst fluorescence was measured in the DAPI channel using the Cytation5 Cell Imaging Multimode Reader (BioTek as part of Agilent, Santa Clara, CA USA) for normalization. Data were processed with the Seahorse Wave Pro Software (version 10.0.1, Agilent, Santa Clara, CA, USA). OCR and ECAR levels are displayed per 1000 cells.

### 2.14. JC-1 Flow Cytometry Analysis

Cells were seeded at a concentration of 6 × 10^5^ cells/well into 6-well plates and allowed to recover overnight. On the next day, cells were treated with the compounds for 24 h. Then, the cell supernatant and trypsinized cells were collected. Cell pellets were washed with PBS and cells were stained with JC-1 solution (1:100 diluted in medium) and incubated for 15 min at 37 °C in the dark. Cells were washed with medium, centrifuged, resuspended in PBS, and analyzed at 530 nm and 605 nm using flow cytometry (LSRFortessa^TM^ X-20 Cell Analyzer, BD Biosciences, Franklin Lakes, NJ, USA). Quantification was performed using FlowJo_v10 software (Becton, Dickinson and Company, Franklin Lakes, NJ, USA)

### 2.15. Statistical Analysis

Statistical analysis was performed using GraphPad Prism version 8.0.1 for Windows (GraphPad Software, San Diego, CA USA).

## 3. Results

### 3.1. Anticancer Activity of cisPt(PhB)_2_ in Platinum-Resistant and -Sensitive Cancer Cells In Vitro

In this study, a broad panel of cancer cell lines of different human or murine origin was used including colorectal, breast, ovarian, and pancreatic carcinoma as well as melanoma and mesothelioma to test the anticancer activity of cisPt(PhB)_2_. Moreover, several cell models with known (platinum) resistance mechanisms were included. IC_50_ values were calculated from viability experiments after 72 h incubation and are depicted in Table 1. In general, while cisplatin was able to inhibit cancer cell growth at low µM concentrations, the cisPt(PhB)_2_ IC_50_ values were mainly in the nM range (Figure 1A,B). Noteworthy, in good agreement with the literature [26,27,28], PhB alone showed only rather low cytotoxicity with IC_50_ values in the mM range (Figure 1B). Across all cell lines tested, cisPt(PhB)_2_ was on average ~36-fold more active compared to cisplatin, paralleled by an enhanced uptake of the complex into the cancer cells (Figure 1C). Both findings are in good agreement with previous results [8]. Of note, while cisplatin had low activity in colorectal cancer cell lines (average IC_50_ value ~8.8 µM), cisPt(PhB)_2_ showed a ~45-fold higher activity in this tumor entity. This is especially interesting as cisplatin typically has reduced efficacy against gastrointestinal cancers in the clinic [29]. A similar increase in activity was also observed in pancreatic cancer cell models. In total, the cell lines most sensitive to cisPt(PhB)_2_ treatment were the human colon carcinoma RKO, the human pancreatic carcinoma Capan-1, and the human ovarian carcinoma cell line A2780.

With regard to drug resistance, two cell models with acquired resistance to cisplatin were investigated: a subclone of the ovarian carcinoma model A2780 and the mesothelioma cell line P31. While in the resistant subclones, A2780/cisR and p31/cisR cisplatin were up to 5.6-fold less effective compared to parental cells, cisPt(PhB)_2_ toxicity was widely unchanged, suggesting that cisPt(PhB)_2_ might not be affected by the same resistance mechanisms as cisplatin [30]. In addition, the TP53 (mutation) status of the cancer cells, which has been associated with intrinsic resistance to platinum drugs [31,32], had no profound impact on the sensitivity of the cells to cisPt(PhB)_2_ in contrast to cisplatin. Thus, the complex is promising for the treatment of cisplatin-resistant cancer types.

### 3.2. Anticancer Activity In Vivo

As a next step, we were interested in the anticancer activity of cisPt(PhB)_2_ in vivo. To this end, two murine cell models (CT26 and B16) were injected subcutaneously as allograft models into immunocompetent mice of two different strains (according to their strain of origin: Balb/c for CT26 and C57BL/6 for B16).

Noteworthy, the C57BL/6 mouse strain in general displays an enhanced sensitivity to platinum drugs, so frequent adaptions in the applied doses are necessary to avoid toxicity. To test cisPt(PhB)_2_, mice received 20 mg/kg per oral gavage, and anticancer efficacy was compared to mice receiving the solvent (Figure 2). In the case of CT26-bearing animals, even repeated administration of the cisPt(PhB)_2_ (three consecutive days for two weeks), only slightly influenced the tumor growth and had no impact on overall survival (Figure 2A). In the case of the second mouse model, comparable to other platinum drugs, the C57BL/6 mouse strain was more sensitive to cisPt(PhB)_2_; thus, only two applications were possible. Nevertheless, two consecutive applications of cisPt(PhB)_2_ were able to significantly stop B16 tumor growth for up to 18 days, resulting in a significantly prolonged overall survival of the animals (Figure 2B). These experiments indicate that B16 tumors are distinctly more sensitive to cisPt(PhB)_2_ than CT26, which prompted us to investigate the cellular and molecular mechanisms underlying the activity of cisPt(PhB)_2_ in these two cell models.

### 3.3. CisPt(PhB)_2_ Displays an Earlier Onset of Cytotoxicity than Cisplatin, Which Is More Pronounced in B16

To further examine the cytotoxic effects of the compounds on CT26 and B16 cells, we performed live-cell microscopy analyses. In more detail, the cells were treated at a concentration of 5 µM and images were taken every 20 min for 24 h (Figure 3A,B). For cisplatin, cells remained viable over the whole imaging period with reduced cell proliferation, especially in the case of B16. This is not unexpected as DNA-targeting compounds typically require a longer incubation period to take a cytotoxic effect, as cell division necessarily results in DNA damage upon DNA platination [33]. Hence, we unexpectedly see the first effects of cisPt(PhB)_2_ treatment already substantially earlier (before the entire replication-dependent cytotoxicity mechanism of cisplatin could have occurred). In more detail, both cell models visibly reacted to cisPt(PhB)_2_ within the first few hours by retracting into roundish morphology. In addition, CT-26 cells strongly reduced their movement (Figure 3C,D), and the induction of cell death was visible after 12 h. In contrast, in B16 cells, cell death was visible already after 2–3 h, which also prohibited the movement analysis in this cell line. Interestingly, the morphological reactions induced by cisPt(PhB)_2_ also differed from the cells upon PhB treatment (Appendix A). This, on the one hand, suggests again, that cisPt(PhB)_2_ might have additional modes of action compared to cisplatin or PhB alone. On the other hand, it supports the hypothesis that there are differences in the sensitivity and possibly cause of cell death in these two cell models.

### 3.4. Differences in Cell Death Induction by cisPt(PhB)_2_ in CT26 and B16 Cells

To gain more insights into the mode of cell death induced by cisPt(PhB)_2_, annexin V/PI stains were performed and measured using flow cytometry at different time points (1–24 h). In good agreement with the live-cell imaging, apoptotic cell death was detected in up to ~80% of the CT26 cells after 16 h (Figure 4A). Unexpectedly, in the B16 model, only ~35% of the cells could be characterized as “apoptotic” or “necrotic” at the 24 h time point (Figure 4B). This was in line with data from the DAPI stains, where only ~10% of the cells displayed apoptotic nuclei in B16 cells 24 h after treatment with cisPt(PhB)_2_ (Figure 4C).

Moreover, in both CT26 and B16, the surviving cell population at the 24 h time point was characterized by a distinct loss of the mitotic cell fraction (Figure 4D). In the case of cisplatin, both cell models responded rather similarly with mild apoptosis induction, enlarged nuclei (Appendix A), and the loss of the mitotic cell population (Figure 4D) indicating activation of the G2-checkpoint and cell cycle arrest. This was further confirmed using the PI stains of ethanol-fixed cells, where 24 h treatment with cisplatin resulted in an enriched S-G2/M (CT26) or G2/M (B16) fraction, respectively (Appendix A). In contrast, for cisPt(PhB)_2_, again differences in the cell populations surviving 24 h treatment were seen between the two cell models. While surviving CT26 cells had distinctly deranged cell cycle distribution, the remaining B16 cells did not vary in their cell cycle distribution from the untreated cells (Appendix A). In conclusion, the very fast cell death induction of cisPt(PhB)_2_ indicates, in contrast to cisplatin, that DNA damage plays only a minor role in the activity of the new complex. In addition, B16 cells are more sensitive to this mode of cytotoxicity than CT26 cells.

### 3.5. CT26 and B16 Cells Differ in Their Metabolism and Albumin Homeostasis

PhB is an aromatic short-chain fatty acid, which, in addition to its ER stress-protecting properties, has also been discussed to impact on the cellular metabolism by direct reaction with CoA, an essential molecule for fatty acid metabolism [34,35]. The most important transport protein of fatty acids in the blood serum is albumin [36]. During our latest studies on the albumin homeostasis of cancer cells [37], we discovered that CT26 cells have a distinctly higher albumin uptake compared to B16 cells (Figure 5A,B). To investigate whether the presence of albumin in the cell supernatant has an impact on the cytotoxicity of cisplatin or cisPt(PhB)_2_, we performed a series of viability assays (Figure 5C–F). The addition of albumin to the cell culture medium resulted in reduced cytotoxicity of both cisplatin as well as cisPt(PhB)_2_. In more detail, cisplatin activity was 2.4-fold and 1.4-fold reduced in CT26 and B16 cells, respectively. In the case of cisPt(PhB)_2_, this effect was much more pronounced in the CT26 cells, where we observed a >5.2-fold reduction in activity upon albumin addition (25 g/L), while only 1.7-fold protection was detected in B16 cells. Consequently, we hypothesized that CT26 and B16 exhibit differences in their metabolism, resulting in a reduced need for fatty acids, and affecting their sensitivity to cisPt(PhB)_2_. According to the literature, CT26 cells display high ATP synthase activity, depend on aerobic respiration, and concomitantly consume high levels of oxygen [26]. In contrast, B16 cells were reported to be characterized by high lactate production [38], suggesting that these cells display “Warburg-like” characteristics [39]. To confirm this, we analyzed the energy metabolism of these models with Seahorse measurements using the glycolytic rate assay as well as the mitochondrial stress test. As shown in Figure 6A,B, CT26 cells had only half the basal lactate production of the B16 cells. Moreover, inhibition of mitochondrial respiration by rotenone/antimycin A (Rot/AA) forced CT26 cells into full glycolysis (max. lactate production), while extracellular acidification in B16 cells was not affected. This demonstrates that the amount of acidification in the B16 cells is mainly due to glycolysis and not due to mitochondrial CO_2_. Additionally, it indicates that B16 cells already run at the maximal glycolysis rate. Inhibition of glycolysis by exposure to the glucose analog 2-DG confirmed that these processes are glycolysis-dependent. With regard to the mitochondrial activity (Mito Stress Test, Figure 6B,D), CT26 had a 2.3-fold higher OCR than B16 cells, which could be completely inhibited by the ATP synthase inhibitor oligomycin.

Moreover, the addition of the protonophore and uncoupling agent carbonyl cyanide-p-trifluoromethoxyphenylhydrazone (FCCP) revealed that the total mitochondrial respiratory capacity of CT26 cells was more than 2.3-fold higher than in B16 cells. When the cells were treated with 5 µM cisPt(PhB)_2_, only B16 cells were affected in their respiration.

More precisely, cisPt(PhB)_2_ had no effect on CT26 cells, while it reduced the maximal mitochondrial respiration in B16 cells by ~40% (Figure 6 and Appendix A). Since the treatment also could not restore glycolytic capacity by inhibition of mitochondrial respiration via Rot/AA (but instead reduced the lactate release in B16 by ~25%), a general inhibition of the B16 cells respiration capacity seems likely, rendering them sensitive towards the metabolic effect of cisPt(PhB)_2_.

To further investigate the metabolic differences with regard to cisPt(PhB)_2_ activity, we decided to examine the mitochondrial membrane potential (ΔΨ) using JC-1 stains. This cationic carbocyanide dye accumulates in mitochondria and exists as monomers or aggregates, which alters the emission spectrum in accordance to the mitochondrial ΔΨ [40]. While CT26 cells harbor normal mitochondrial function, in the B16 line, a higher fraction of cells already exhibits low mitochondrial ΔΨ in untreated conditions (Figure 6E).

This is in good agreement with their mitochondrial deficiency indicated by the Seahorse experiments (compare Figure 6A–D). Upon cisPt(PhB)_2_ treatment (24 h), CT26 cells displayed a significantly elevated level of depolarized mitochondria (Figure 6F). In contrast, the mitochondrial ΔΨ of B16 cells remained largely unaffected upon treatment with cisplatin or cisPt(PhB)_2_. Together with the data above, this indicates that cells with normal respiratory capacities such as CT26 (or MCF-7 [8]) die from cisPt(PhB)_2_-induced apoptosis via the intrinsic mitochondrial pathway, while this process is not activated in “Warburg-like” cells such as B16.

### 3.6. CisPt(PhB)_2_ Activity Is Associated with Enhanced Lipid Droplet Formation

To investigate the impact of cisPt(PhB)_2_ on fatty acid metabolism, we performed live-cell microscopic analysis using bodipy™ 493/503 as a marker for neutral lipid as present in lipid droplets [41]. Generally, under control conditions, CT26 cells displayed 10-fold higher basal levels of lipid droplets than B16 cells (Figure 7A,B). Treatment with cisPt(PhB)_2_ induced lipid droplet accumulation in both cell lines. Notably, this effect was much more pronounced in B16 (13-fold increase in bodipy™ 493/503 foci compared to the basal levels) than in CT26 cells (2.2-fold increase), shifting cisPt(PhB)_2_-treated B16 cells into lipid droplet concentrations comparable to untreated CT26. To gain more insight into the role of lipid droplets in cisPt(PhB)_2_ activity, we co-treated the cells with the non-specific long-chain acyl-CoA synthetase inhibitor triacsin C. The compound prevents formation of acyl fatty acids, which are the building blocks of lipid droplets and, thus, is supposed to inhibit fatty acid-induced apoptosis (lipoapoptosis) [42]. The combination treatment for 24 h had strong antagonistic effects in the case of CT26 but not in B16 cells (Figure 7C,D), suggesting that, in fact, lipid droplet formation supports cisPt(PhB)_2_ activity. As for catabolism, fatty acids need to be transported across the mitochondrial membrane by palmitoyltransferases (CPTs) [43]; thus, we investigated the impact of the CPT inhibitors etomoxir or perhexiline. Noteworthy, preliminary Seahorse experiments with etomoxir alone indicated a stronger dependence of CT26 as compared to B16 cells on mitochondrial fatty acid catabolism (Figure 7E). Thus, it was rather unexpected that, especially in the B16 cells, a visible antagonism of etomoxir with cisPt(PhB)_2_ was observed (while the compounds were additive in CT26 cells). In contrast, perhexiline had weakly antagonistic effects to a similar extent in both cell models.

Taken together, these data suggest that cellular lipid metabolism plays a role in cisPt(PhB)_2_ activity and that cells with “Warburg-like” phenotype (characterized by reduced and abnormal mitochondrial activity and enhanced aerobic glycolysis) are more vulnerable to this complex than cells with “normal” metabolism.

## 4. Discussion

Platinum-based drugs are among the most frequently used anticancer agents, especially at the late stage of the disease. However, drug resistance, for example, based on reduced drug uptake (e.g., downregulation of CTR1), changed damage recognition (e.g., TP53 mutation), or enhanced DNA damage repair (e.g., via DNA excision repair) distinctly hampers successful therapy [32]. Consequently, compounds with improved anticancer efficacy and altered activity profiles are of central interest. Here, drugs which exploit differences between healthy and malignant cells are especially promising because they allow more selective targeting of the cancer tissue. In this study, we investigated the mode of action of a novel dual-action platinum(IV) complex, cisPt(PhB)_2_, which releases upon activation two clinically used drugs: cisplatin and PhB [8]. PhB is an interesting combination partner for cisplatin, as there are several reports on their synergistic mode of action [13,44]. However, PhB is limited in its activity, and due to its negative charge at physiological pH, its passage through the cell membrane is very limited. This is also reflected by the very high IC_50_ values of the compound (in the mM range) in cell culture studies [26,27,28]. CisPt(PhB)_2_ facilitates synergistic accumulation of both cisplatin and PhB. On the one hand, PhB increases the lipophilicity compared to cisplatin, hence facilitating enhanced uptake of platinum. On the other hand, it neutralizes the negative charge of PhB, enhancing PhB accumulation. Indeed, cisPt(PhB)_2_ was highly active in the nM range against a broad panel of cancer cell models and is able to circumvent the most common resistance mechanisms against platinum drugs. This is exciting as cisPt(PhB)_2_ is distinctly more active in vitro than the two “parent compounds”. These data are also in good agreement with the work of Raveendran et al. [8], who already reported that, although cisPt(PhB)_2_ induced the stabilization of TP53 in MCF-7 cells, the TP53 knock-out subclone of HCT116 showed similar sensitivity to cisPt(PhB)_2_ as the parental cells. This suggests that, contrary to cisplatin, cisPt(PhB)_2_ does not depend on TP53 (a mediator for DNA damage response) for its activity. This is rather surprising since, according to reports in the literature, the anticancer activity of a set of HDAC inhibitors (including PhB) is at least in part based on TP53 [45,46]. In addition, more recently, Romeo et al. reported a selective sensitivity of glioma cell lines carrying a mutated version of TP53 to PhB [26]. Thus, our viability data suggested that cisPt(PhB)_2_ might have modes of action other than the so far designated DNA-targeting/HDAC-inhibiting properties [8].

In addition to the cell culture analysis, the first in vivo experiments using immune-competent allograft models suggested that cisPt(PhB)_2_ can be successfully applied orally with promising activity especially against B16 melanoma cells. Noteworthy, B16 cells are of the so-called “Warburg” phenotype [47,48], which summarizes their distinct metabolic state differently from healthy tissue. Warburg-like cells are characterized by abnormal mitochondrial activity, which pushes the cells towards aerobic glycolysis [47,48]. As a consequence, these cells have a reduced uptake of the (fatty acid) carrier serum protein albumin together with distinct differences in fatty acid metabolism (compare Figure 5A) [49]. Our data further suggest that Warburg-like cells are less capable of coping with cisPt(PhB)_2_, leading to the rapid induction of a non-apoptotic form of cell death. In contrast, in cancer cells with healthy mitochondria (e.g., CT26 and MCF-7 [8]), apoptosis via the mitochondrial pathway was seen. Noteworthy, cisPt(PhB)_2_ distinctly differed in its mode of action from cisplatin, and the rapid cell death induction (within a few hours) especially suggested that DNA damage is not involved in the first stage of its activity. Thus, we expect that the cisplatin arm of the mode of action will affect mainly the cell fraction surviving the first cytotoxic phase of drug activity (still, this functionality of the complex might be effective in killing potential residual cell clones later on). However, whether the observed anticancer activity originates solely from the released PhB is difficult to answer. A big obstacle in the investigation of this question is the ~10.000-fold difference in IC_50_ values between cisPt(PhB)_2_ and PhB. Moreover, it is unclear how the potential differences in drug delivery impact the intracellular PhB distribution, which allows for new or more pronounced (PhB-associated) biological activities.

In general, PhB is a drug with multiple modes of action. It is, for example, a chemical chaperon, stabilizing protein conformation, and, thus, one of the most frequently used ER stress inhibitors [15,16,17]. However, to the best of our knowledge, the exact mechanism underlying this effect is not fully understood. What is known is that PhB influences cellular metabolism by binding to CoA via thiol adduct formation [18] and by inhibiting histone deacetylase (HDAC) [19]. Moreover, PhB is able to directly inhibit mitochondrial PDK1 [20] as well as β-oxidation in the mitochondria in a competitive manner [34]. Interestingly, although PhB is approved for the treatment of certain metabolism-associated diseases (e.g., urea cycle disorders), the impact of this drug with respect to typical cancer-associated metabolic changes such as the Warburg effect is widely unexplored. This is surprising considering that the difference in metabolic properties between healthy and malignant tissues provides a promising Achilles heel for tumor-specific cancer treatment. Our data suggest that Warburg-like cells might be hypersensitive to the cellular PhB delivery by cisPt(PhB)_2_ and respond with non-apoptotic cell death induction. In more detail, B16 cells showed reduced fatty acid uptake via albumin and thus lower cellular lipid pools (10-fold fewer lipid droplets compared to CT26 cells). This can be explained by the fact that mitochondrial respiration in cancer cells usually is more dependent on fatty acid metabolism than the glucose pathway [49]. CisPt(PhB)_2_ seems to impact these pathways, stimulating the formation of lipid droplets. Moreover, inhibitors that interfere with cellular fatty acid metabolism such as triascin C (inhibitor of the acyl CoA synthetase and thus prevents formation of lipid droplets [50]), or etomoxir and perhexilin (both inhibiting the transport of fatty acids into the mitochondria [51,52]), affected cisPt(PhB)_2_ activity. Interestingly, co-treatment of cisPt(PhB)_2_ with triascin C only led to an antagonistic effect in CT26 cells. This suggests that in CT26 cells, cell death is mediated by cytotoxic lipid intermediates via the intrinsic mitochondrial pathway (Figure 8). However, this also allows these cells to activate anti-apoptotic signaling pathways in the mitochondria [53]. Consequently, the mitochondria with their specific fatty acid metabolism and apoptosis-regulating function could serve as a buffer against cisPt(PhB)_2_-induced effects. In contrast, this option is limited in “Warburg-like” cells such as B16, forcing them to die via a yet undefined, potentially lipid-associated alternative pathway. In this context, the difference between etomoxir and perhexiline is noteworthy because the activity of both compounds has been connected to the transport of fatty acids into the mitochondria. Etomoxir displayed antagonistic activity mainly in B16 cells (while perhexiline had similar activity in CT26 and B16 cells). This effect can be explained by an additional etomoxir mode of action, namely the binding to CoA at higher concentrations [54], a functionality shared with PhB [18]. Due to their reduced dependency on fatty acids for mitochondrial respiration, B16 cells are assumed to produce all of their acetyl-CoA (needed for protein acetylation) via glycolysis-derived pyruvate. This hypothesis would be in line with the inhibition of PDK1 through PhB, leading to an enhanced activation of the pyruvate dehydrogenase that converts pyruvate to acetyl-CoA. This could render these cells more sensitive to the CoA-binding of PhB. Nevertheless, the exact mechanism(s) underlying the effects observed with cisPt(PhB)_2_ definitely warrant more in-depth investigations, and further studies are required to analyze the exact impact of cisPt(PhB)_2_ on cancer cell metabolism. This is especially of interest considering several reports that drug-resistant cancer cells are characterized by an altered metabolism [32].

In summary, cisPt(PhB)_2_ is a novel orally active anticancer compound using a cisplatin-releasing platinum(IV) platform for the improved delivery of PhB into cancer cells. This results in enhanced anticancer activity against metabolically altered cancer cells in vivo. Therefore, cisPt(PhB)_2_ is an interesting candidate for further preclinical investigations.

## Figures and Tables

**Figure 1 pharmaceutics-15-00677-f001:**
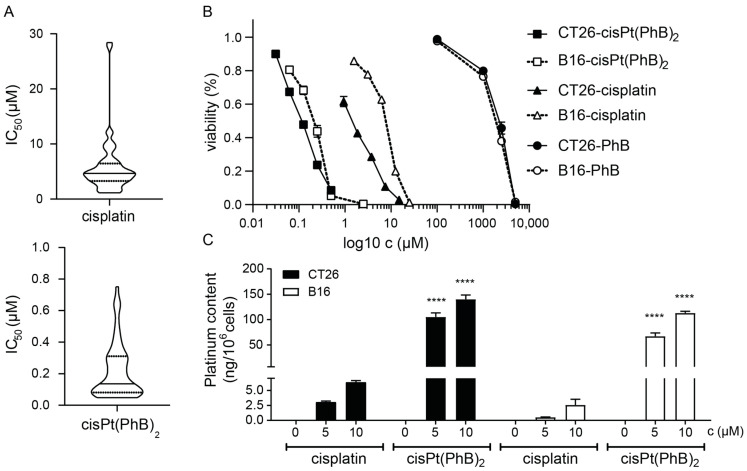
**Anticancer activity and drug uptake of cisPt(PhB)_2_ compared to cisplatin.** (**A**) Violin plots of the IC_50_ values from cisPt(PhB)_2_ or cisplatin, respectively (without cisplatin resistant cell models). (**B**) Dose–response curve of the indicated compounds in CT26 and B16 cells after 72 h incubation. Cell viability was detected using MTT assays. Values are given as mean ± SD from triplicates. (**C**) Cellular platinum levels after 2 h treatment with equimolar concentrations of cisplatin or cisPt(PhB)_2_ were measured using ICP-MS. Values were normalized to cell number. Bars depict mean ± SD from three replicates. Significance compared to respective cisplatin-treated samples was calculated using two-way ANOVA and Tukey’s multiple comparisons test; **** *p* < 0.0001.

**Figure 2 pharmaceutics-15-00677-f002:**
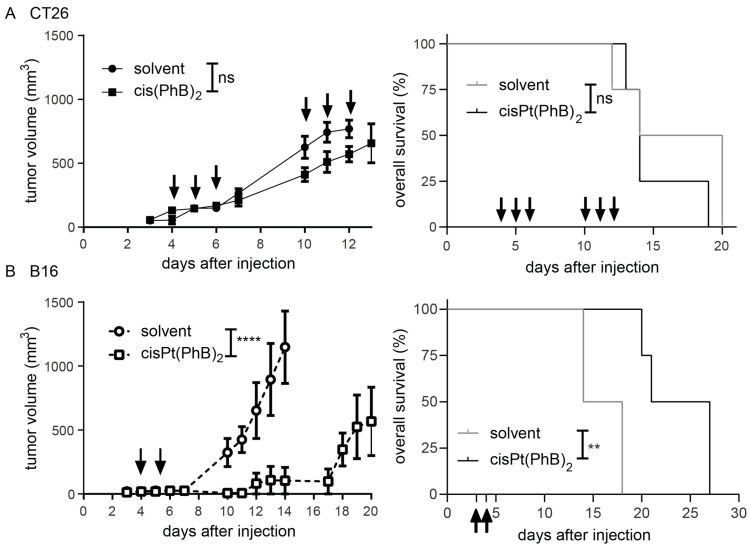
**Anticancer activity in vivo**. Left graphs show tumor growth of CT26-bearing Balb/c mice or B16-bearing C57BL/6 mice that were treated with (**A**) cisPt(PhB)_2_ (20 mg/kg, p.o.) on day 3, 4, 5, 10, 11, and 12, or (**B**) cisPt(PhB)_2_ (20 mg/kg, p.o.) on day 3 and 4 after tumor cell injection. Significance was calculated using two-way ANOVA. ns—non-significant, ** *p* < 0.01, **** *p* < 0.0001. Right graphs depict Kaplan–Meier curves showing the survival of mice. Significance was calculated using Mantel–Cox test ns—non-significant, ** *p* < 0.01.

**Figure 3 pharmaceutics-15-00677-f003:**
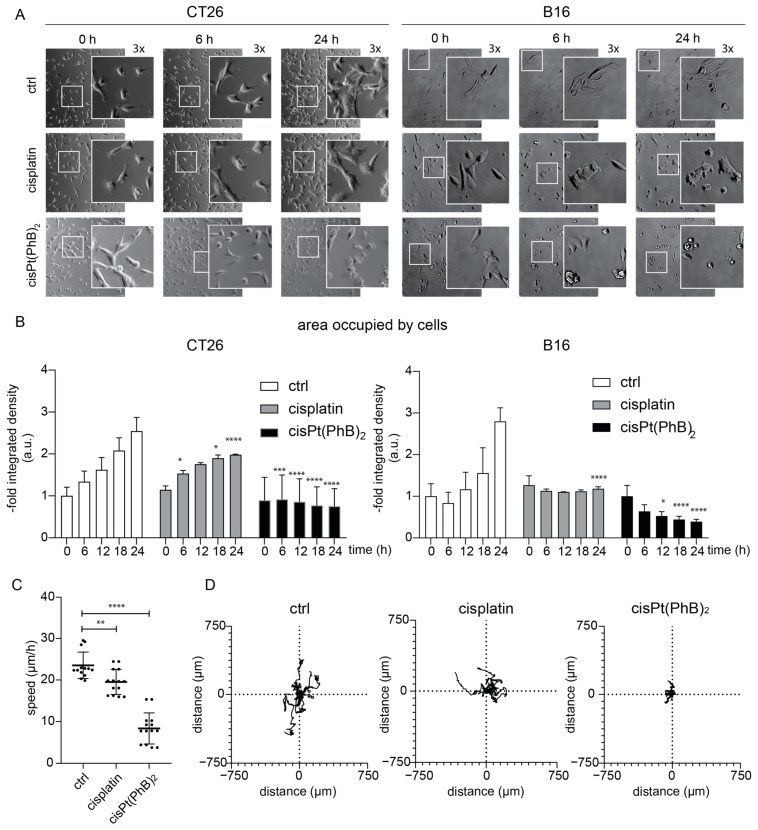
**CisPt(PhB)_2_ has a faster anticancer activity than cisplatin.** (**A**) Live-cell images show CT26 and B16 cells treated with 5 µM cisplatin or cisPt(PhB)_2_ after various time points (20× and additional 1.5× magnification). (**B**) The graph shows the quantification of the area occupied by cells from (**A**). Bars depict mean ± SD from two representative images (683 × 683 µm), and statistical significance compared to control values was calculated using two-way ANOVA with Sidak’s multiple comparison test; * *p* < 0.05, *** *p* < 0.001, **** *p* < 0.0001. (**C**) Graphs show the speed (µm/h) of movement of depicted cells from CT26 cells of (**A**). Significance was calculated using two-way ANOVA and Sidak’s multiple comparison test; ** *p* < 0.01, **** *p* < 0.0001. (**D**) The graph shows the movement of ten individual cells of CT26 cells of (**A**).

**Figure 4 pharmaceutics-15-00677-f004:**
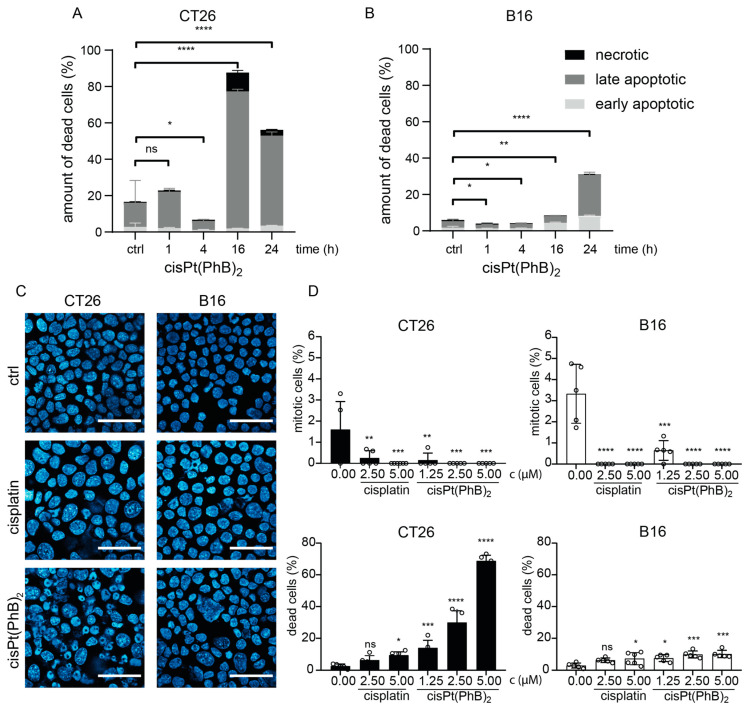
**CisPt(PhB)_2_ exhibits distinct anticancer activity in CT26 or B16 cells.** Quantification of annexin-V/PI stains using flow cytometry of CT26 (**A**) or B16 (**B**) cells treated with 5 µM cisPt(PhB)_2_ for 1, 4, 16, or 24 h. Statistical significance compared to control values was calculated using ordinary one-way ANOVA and Holm–Sidak’s multiple comparisons test; ns—non-significant, * *p* < 0.05, ** *p* < 0.01, **** *p* < 0.0001. (**C**) DAPI-stained nuclei of aceton/methanol-fixed CT26 and B16 cells after treatment with 5 µM cisplatin or cisPt(PhB)_2_ for 24 h. (**D**) shows quantification of mitotic or apoptotic nuclei from (**C**), respectively. Statistical significance compared to control values was calculated using one-way ANOVA and Dunnett’s multiple comparisons test; ns—non-significant, * *p* < 0.05, ** *p* < 0.01, *** *p* < 0.001, **** *p* < 0.0001.

**Figure 5 pharmaceutics-15-00677-f005:**
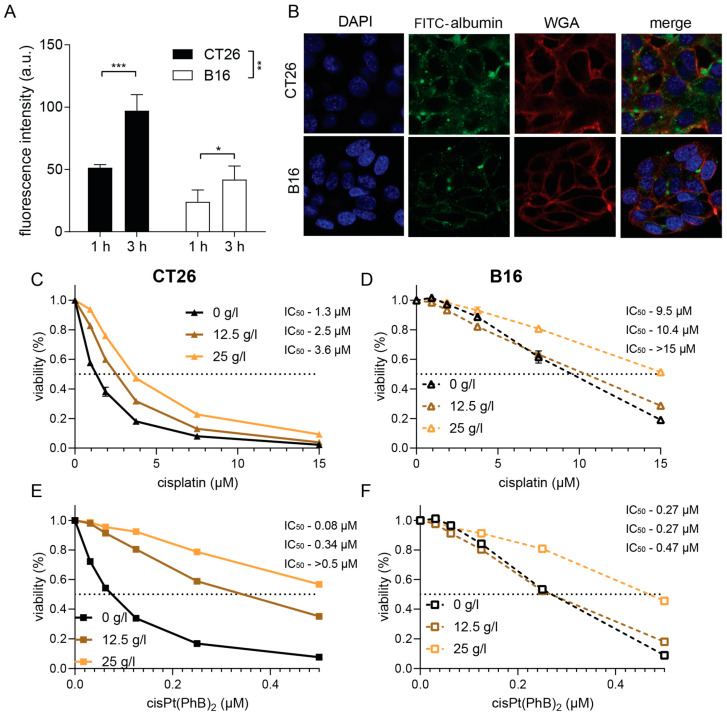
**CT26 and B16 cells display different albumin uptake efficacy protecting from cisPt(PhB)_2_ cytotoxicity.** (**A**) Cellular uptake of FITC-labelled albumin in CT26 and B16, respectively. Bars depict mean ± SD from three independent experiments. Significance was calculated using two-way ANOVA and Dunnett’s Multiple Comparison test; * *p* < 0.05, ** *p* < 0.01, *** *p* < 0.001. (**B**) Representative fluorescence images of FITC-albumin uptake (green) in CT26 or B16 cells stained for nuclei (DAPI, blue) and membranes (WGA, red) after 3 h incubation with FITC-labelled albumin (green). (**C**–**F**) Graphs show viability of (**C**,**E**) CT26 or (**D**,**F**) B16 cells treated in serum-free medium with cisplatin or cisPt(PhB)_2_ in the absence or presence of albumin (12.5 or 25 g/L) for 72 h. Cell viability was detected using MTT assays. Values are given as mean ± SD from triplicates.

**Figure 6 pharmaceutics-15-00677-f006:**
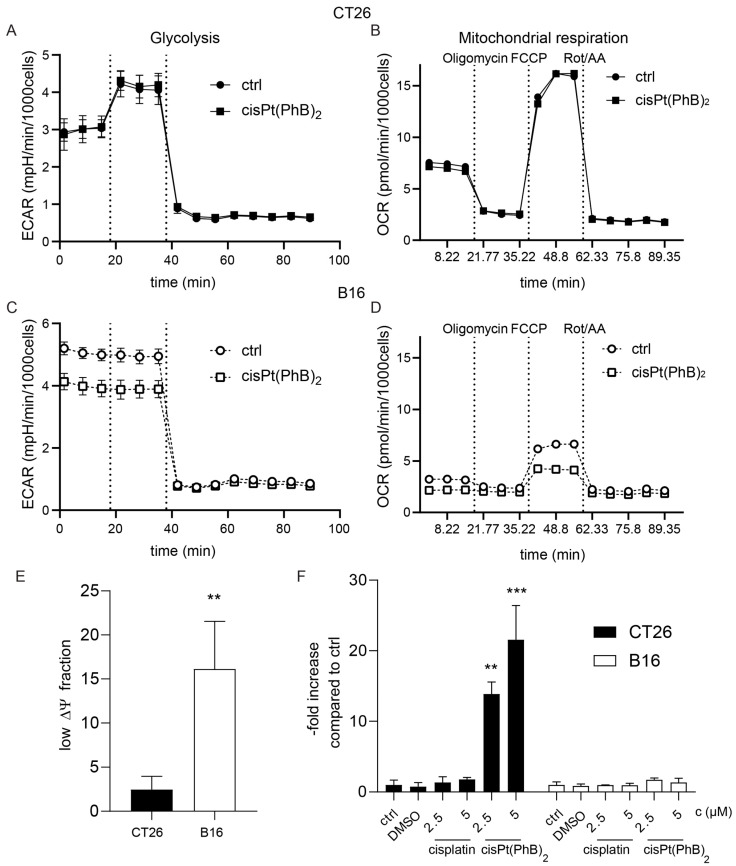
**CT26 and B16 cells display different energy metabolism.** Graphs show (**A**,**C**) Extracellular acidification rate (ECAR) in a glycolytic rate assay and (**B**,**D**) oxygen consumption rate (OCR) in a mitochondria stress test of CT26 and B16 cells, respectively, measured with a real-time Seahorse experiment under basal conditions and in response to indicated mitochondrial inhibitors. Cells are either untreated or pretreated for 4 h with 5 µM of cisPt(PhB)_2_. (**E**) Fraction of cells with low mitochondrial ΔΨ. CT26 and B16 cells were stained with JC-1 followed by flow cytometry measurement. (**F**) Graphs depict fold increase of low mitochondrial ΔΨ in CT26 and B16 cells compared to control values after 24 h treatment with the indicated compounds. Significance was calculated using one-way ANOVA and Dunnett’s multiple comparisons test; ** *p* < 0.01, *** *p* < 0.001.

**Figure 7 pharmaceutics-15-00677-f007:**
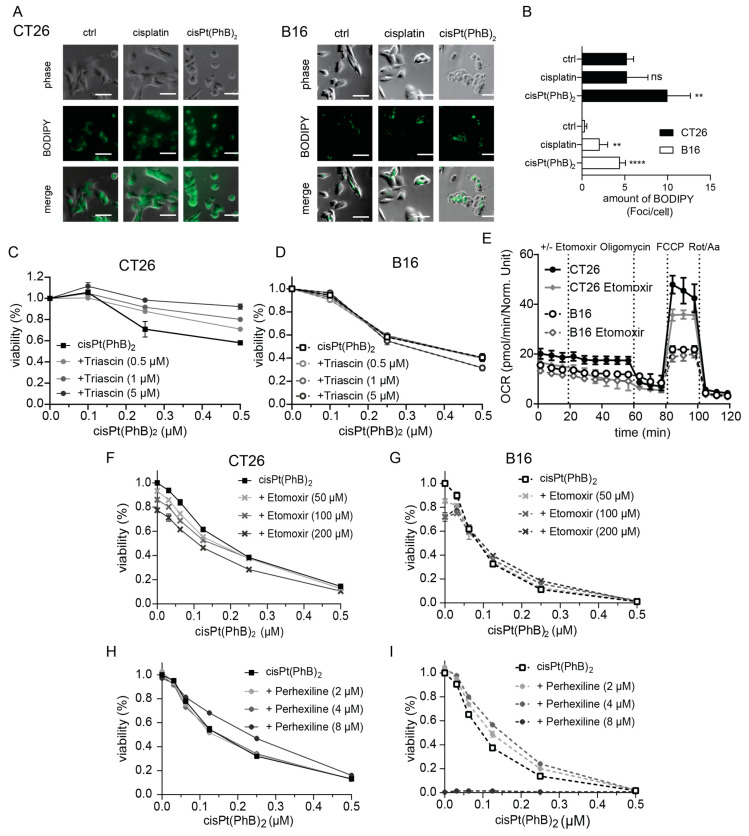
**CisPt(PhB)_2_ affects the cellular lipid metabolism.** (**A**) Cells were stained with BODIPY™ 493/503 and treated with 5 µM cisplatin or cisPt(PhB)_2_ for 10 h. Images (bright field and GFP channel) were taken every 10 min for 25 h. (**B**) Graph shows quantification of BODIPY™ 493/503 foci from (**A**) after 10 h treatment. Significance was calculated using one-way ANOVA and Dunnett’s multiple comparisons test; ns—non-significant, ** *p* < 0.01, **** *p* < 0.0001. (**C**,**D**) Cytotoxicity assays of combination treatment with triacsin C and cisPt(PhB)_2_ in (**C**) CT26 and (**D**) B16 cells after 24 h. (**E**) OCR in a substrate oxidation stress test with etomoxir (4 µM) of CT26 and B16 cells measured in real-time seahorse experiments under basal conditions. (**F**–**I**) Cytotoxicity assays of combination treatment with etomoxir (**F**,**G**) as well as perhexiline (**H**,**I**) and cisPt(PhB)_2_ in CT26 and B16 cells after 72 h. Cell viability was detected using MTT assays. Values are given as mean ± SD from triplicates.

**Figure 8 pharmaceutics-15-00677-f008:**
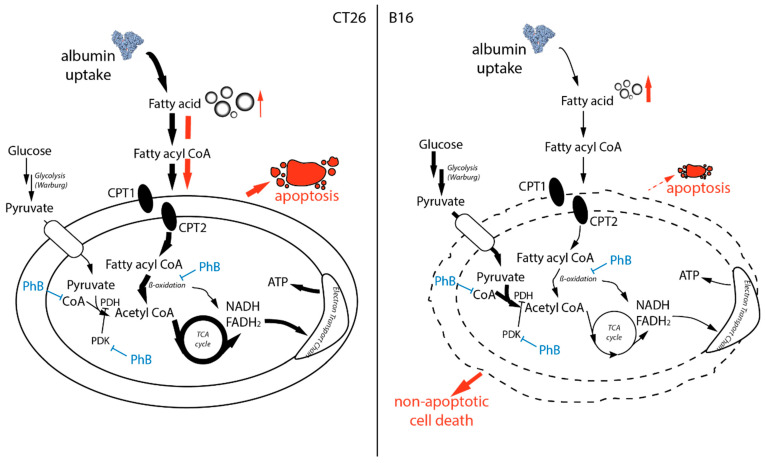
Hypothetical mode of action of cisPt(PhB)_2_ in CT26 vs. B16 cells. Cartoons depict the mitochondria and their cytosolic surrounding. Bold arrows indicate metabolic pathways that are enhanced in the specific cell types. Red indicates the observed effects after treatment with cisPt(PhB)_2_. Blue indicates the mode of action of PhB alone according to the literature.

**Table 1 pharmaceutics-15-00677-t001:** Cytotoxicity as IC_50_ values are depicted as mean ± standard deviation (SD) from at least two independent experiments. Ratios of the IC_50_ values from cisplatin: cisPt(PhB)_2_ and of p53-knock out (KO) or cisplatin-resistant (cisR) cells compared to the respective parental cell line were calculated.

Tumor Entities	IC_50_ (µM) Mean ± SD				Resistance Ratio	
	Cell Line	Cisplatin	cis(PhB)_2_	Ratio	Cisplatin	cisPt(PhB)_2_
colorectal	HCT116	5.8 ± 0.9	0.08 ± 0.01	71.5	2.2	1.8
	HCT116p53KO	13.0 ± 2.6	0.14 ± 0.00	91.9		
	RKO	1.5 ± 0.0	0.06 ± 0.00	25.4	4.6	1.1
	RKOp53KO	7.0 ± 0.0	0.06 ± 0.00	108.6		
	SW480	7.1 ± 0.4	0.10 ± 0.00	74.0		
	CT26 *	1.4 ± 0.2	0.13 ± 0.03	10.8		
	MC38 *	28.0 ± 0.6	0.60 ± 0.08	46.6		
ovarian	A2780	1.6 ± 0.6	0.07 ± 0.01	22.1	5.7	1.2
	A2780/cisR	9.0 ± 0.8	0.09 ± 0.01	101.8		
pancreas	SKOV3	3.7 ± 1.0	0.34 ± 0.07	10.9		
	Capan-1	3.7 ± 1.2	0.06 ± 0.01	60.7		
	PANC-1	10.7 ± 1.6	0.25 ± 0.04	42.9		
melanoma	VM1	4.0 ± 0.7	0.22 ± 0.09	18.1		
	B16 *	8.6 ± 1.4	0.24 ± 0.06	35.7		
mesothelium	p31	9.7 ± 0.7	0.57 ± 0.10	17.0	3.2	1.4
	p31/cisR	30.9 ± 8.6	0.80 ± 0.14	38.6		
breast	MDA-MB-231	4.9 ± 0.7	0.17 ± 0.03	28.7		

* indicates mouse as species of origin.

## Data Availability

Data is contained within the article or Appendix A. Additional information is provided upon request.

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
