# Peer review of "Influence of the Fatty Acid Metabolism on the Mode of Action of a Cisplatin(IV) Complex with Phenylbutyrate as Axial Ligands"

_pharmaceutics, 2023, doi:10.3390/pharmaceutics15020677_

Round 1

Reviewer 1 Report

Manuscript ID: pharmaceutics-2203505

Title: Influence of the fatty acid metabolism on the mode of action of a  novel cisplatin complex(IV) with phenylbutyrate as axial ligands Authors: Theresa Mendrina, Isabella Poetsch, Hemma Schueffl, Dina Baier,  Christine Pirker, Alexander Ries, Bernhard K. Keppler, Christian R. Kowol, Dan Gibson, Michael Grusch, Walter Berger, Petra Heffeter *

The manuscript submitted by Mendrina et al. reports an in-depth investigation of the anticancer properties of cisPt(PhB)2 in cell culture and mouse allograft experiments. The authors demonstrated improved anticancer activity of the complex compared with cisplatin and PhB alone. They also reported that the cellular fatty acid metabolism and mitochondrial activity distinctly impacted the compound´s mode of action. The “Warburg-like” cells, which are characterized by deficient mitochondrial function and fatty acid catabolism, are less capable of coping with cisPt(PhB)2 leading to rapid induction of a non-apoptotic form of cell death. The authors further concluded that cisPt(PhB)2 has a promising activity, especially against cisplatin-resistant cancer cells with “Warburg-like” properties.

I think that the manuscript contains well-performed experiments supporting the scientific conclusions. Several minor concerns need to be addressed before the acceptance of the manuscript.

1.       I do not think that it is proper to call investigated complex a drug.

2.       The part of the manuscript starting with Overall, these data…Line 287 up to Line 297 located in Results is a part of the Discussion.

3.       Figure 1B…c…concentrations…this is the only Figure where X axis is named in this way.

4.       Figure 2A in legend has 0.9% NaCl. Figure A right and Figure 2B have solvent. Is there a difference?

5.       Figure 3D…what does represent axis X and what axis Y?

6.       How are you explaining the drop in cell number in CT26 Figure 4A?

7.       Figure 4D is expanded Figure 4C. I do not see the reason for indicating it as C and D. Figure 4D and E..axis X.. numbers to one decimal place!

8.       In general, all Figures look very mismatched. They are not aligned concerning size, X and Y axis naming (axis names sometimes start with a capital letter, sometimes with lowercase, sometimes with a unit). In some figures, for example, the time shown on the axis is next to the numbers, in some places it is not. That bothers me a lot. The Figures look like they were made by many people, each in their way, and then just stitched together.

9.       Figure 5C-F 25 g/L is too bright.

Author Response

The manuscript submitted by Mendrina et al. reports an in-depth investigation of the anticancer properties of cisPt(PhB)2 in cell culture and mouse allograft experiments. The authors demonstrated improved anticancer activity of the complex compared with cisplatin and PhB alone. They also reported that the cellular fatty acid metabolism and mitochondrial activity distinctly impacted the compound´s mode of action. The “Warburg-like” cells, which are characterized by deficient mitochondrial function and fatty acid catabolism, are less capable of coping with cisPt(PhB)2 leading to rapid induction of a non-apoptotic form of cell death. The authors further concluded that cisPt(PhB)2 has a promising activity, especially against cisplatin-resistant cancer cells with “Warburg-like” properties.

 I think that the manuscript contains well-performed experiments supporting the scientific conclusions. Several minor concerns need to be addressed before the acceptance of the manuscript.

  1. I do not think that it is proper to call investigated complex a drug.

 Following the reviewer´s suggestion, “drug” has been replaced by compound or complex where appropriate.

  1. The part of the manuscript starting with Overall, these data…Line 287 up to Line 297 located in Results is a part of the Discussion.

 As suggested by the reviewer, this section has been moved to the discussion area.

  1. Figure 1B…c…concentrations…this is the only Figure where X axis is named in this way.

 Thank for these suggestions! The Figure has been adapted.

  1. Figure 2A in legend has 0.9% NaCl. Figure A right and Figure 2B have solvent. Is there a difference?

 Thank you for making us aware of these inconsistencies. The solvent was always 0.9% NaCl, which is now also indicated in all Figures.

  1. Figure 3D…what does represent axis X and what axis Y?

 Thank you for making us aware of these missing axis labelings. They have now been included into the Figure!

  1. How are you explaining the drop in cell number in CT26 Figure 4A?

 As the drop in number of dead cells is mainly due to a reduction of the necrotic cells or cells in late-stage apoptosis, one explanation could be that the cell death is already so far progressed that the cell structure is intact anymore. Thus, these cells break apart into tiny pieces, which are subsequently lost during the preparation steps.

  1. Figure 4D is expanded Figure 4C. I do not see the reason for indicating it as C and D. Figure 4D and E..axis X.. numbers to one decimal place!

We have now reduced the subfigure labelings in Figure 4 to A-D. However, we did not change the x axis numbers to one decimal because then we would not be able to indicate the 1.25 µM.

  1. In general, all Figures look very mismatched. They are not aligned concerning size, X and Y axis naming (axis names sometimes start with a capital letter, sometimes with lowercase, sometimes with a unit). In some figures, for example, the time shown on the axis is next to the numbers, in some places it is not. That bothers me a lot. The Figures look like they were made by many people, each in their way, and then just stitched together.

You are correct, there have been several people working on the manuscript. We have now homogenized the formating of all Figures in a more precise manner.  

  1. Figure 5C-F 25 g/L is too bright.

Has been corrected.

Reviewer 2 Report

There are a few minor typographical errors which I have annotated in the PDF.

My only scientific questions is a clarification of the statement in the conclusion that “cisPt(PhB)2 is a novel orally active anticancer compound” it was not clear to me if this was proven through animal experiments or that the combination should be or is hypothesized orally active.

Author Response

There are a few minor typographical errors which I have annotated in the PDF.

 Thank you for indicating these mistakes. They have been corrected!

My only scientific questions is a clarification of the statement in the conclusion that “cisPt(PhB)2 is a novel orally active anticancer compound” it was not clear to me if this was proven through animal experiments or that the combination should be or is hypothesized orally active.

CisPt(PhB)2 has been applied orally in all experiments shown in this manuscript. This is also indicated in the Figure Legends and the Material & Method Section.

Reviewer 3 Report

The work submitted by Petra Heffeter et al. describes a detailed study of the cytotoxic activity and mechanisms of action of a Pt(IV) prodrug able to release the classical cisplatin and phenylbutyrate drugs inside cancer cells. The reported preclinical evaluation included in vitro studies in an enlarged panel of tumor cells, as well as therapeutic assays in murine tumor models. Interestingly, the results showed that the activity and possible mechanisms of action of the Pt(IV) prodrug are different from cisplatin, and are dependent on the metabolic fingerprint of the cells. The work has been performed properly and has been presented in a clear fashion and with scientific rigor. Thus, I am glad to recommend its publication after the authors address the following minor issues:

11)       Page 1, Line 3, Title: I suggest that the authors remove “novel” from the title. In fact, the tested Pt(IV) prodrug is not novel.

22)    Page 2, Line 56: The authors should mention here some recent and relevant references on the same type of approach for other Pt(IV) prodrugs.

33)      Page 2, Line 68: Modify the writing. It is not clear what means “Here”. Do you want to mention the “dual action”?

44)       Page 2, Line 88: Remove the “dd” abbreviation and write it in extent.

55)    Page 6, Line 278: Remove this text.

66)    Page 8, Line 308: Discuss why murine tumor models were used instead of xenograft models of human cancers.

77)     Page 9, Line 327: How does the in vivo antitumoral effects of cisPt(PhB)2 compare with cisplatin in these murine tumor models?

Author Response

The work submitted by Petra Heffeter et al. describes a detailed study of the cytotoxic activity and mechanisms of action of a Pt(IV) prodrug able to release the classical cisplatin and phenylbutyrate drugs inside cancer cells. The reported preclinical evaluation included in vitro studies in an enlarged panel of tumor cells, as well as therapeutic assays in murine tumor models. Interestingly, the results showed that the activity and possible mechanisms of action of the Pt(IV) prodrug are different from cisplatin, and are dependent on the metabolic fingerprint of the cells. The work has been performed properly and has been presented in a clear fashion and with scientific rigor. Thus, I am glad to recommend its publication after the authors address the following minor issues:

 1)       Page 1, Line 3, Title: I suggest that the authors remove “novel” from the title. In fact, the tested Pt(IV) prodrug is not novel.

Following this reviewer´s suggestions, “novel” has been deleted from the title.

 2)    Page 2, Line 56: The authors should mention here some recent and relevant references on the same type of approach for other Pt(IV) prodrugs.

We have now included several new references into the manuscript.

 3)      Page 2, Line 68: Modify the writing. It is not clear what means “Here”. Do you want to mention the “dual action”?

Sorry for being unclear. We have modified the respective sentence.

4)       Page 2, Line 88: Remove the “dd” abbreviation and write it in extent.

 Has been done!

5)    Page 6, Line 278: Remove this text.

 Sorry, for this very embarrassing mistake. The text has been deleted.

6)    Page 8, Line 308: Discuss why murine tumor models were used instead of xenograft models of human cancers.

 As mentioned in the results section, we usually start our experiments with allograft models because we are always interested in the role of immune cells in the anticancer activity (and toxicity) of metal drugs. A very prominent example is oxaliplatin, which (being an immunogenic cell death inducer) is in fact very dependent on the immune system end exerts only low to no anticancer activity in PDX models (compare also Englinger et al Chem Rev. 2019). Of course in order to find the ideal target tumor entities for further (pre)clinical development, it will be crucial and very interesting to investigate the activity of cisPt(PhB)2 also in xenograft models.

7)     Page 9, Line 327: How does the in vivo antitumoral effects of cisPt(PhB)2 compare with cisplatin in these murine tumor models?

In general, the direct comparison of our data on cisplatin and cisPt(PhB)2 is a bit difficult because they are applied via different routes and application schemes (cisplatin i.p. or i.v. in split dose every 4. or 7. day) and cisPt(PhB)2 orally on consecutive days). Moreover, we have so far not compared them back-to-back in one experiment. Nevertheless, it seems that cisPt(PhB)2 is more effective especially against B16 cells, but this has to further investigated in additional experiments.